# Comparative Performance Evaluation of Continuous Monitoring Blood Culture Systems Using Simulated Septic Specimen

**DOI:** 10.3390/diagnostics15040468

**Published:** 2025-02-14

**Authors:** Kwangjin Ahn, Taesic Lee, Sangwon Hwang, Dong Min Seo, Young Uh

**Affiliations:** 1Department of Laboratory Medicine, Yonsei University Wonju College of Medicine, Wonju-si 26426, Republic of Korea; kjahn123@yonsei.ac.kr; 2Department of Family Medicine, Yonsei University Wonju College of Medicine, Wonju-si 26426, Republic of Korea; ddasic123@yonsei.ac.kr; 3Department of Precision Medicine, Yonsei University Wonju College of Medicine, Wonju-si 26426, Republic of Korea; arsenal@yonsei.ac.kr; 4Department of Medical Information, Yonsei University Wonju College of Medicine, Wonju-si 26426, Republic of Korea; dmseo@yonsei.ac.kr

**Keywords:** blood culture, Gram-positive, Gram-negative, *Candida*, obligate anaerobe

## Abstract

**Background/Objectives:** Continuous monitoring blood culture systems (CMBCSs) are revolutionary automated instruments that facilitate the rapid identification of pathogens in blood samples from patients with sepsis. However, with only a few CMBCSs being widely used as references, user dependency on these limited options has grown. In response, a new CMBCS was developed and compared with existing systems to evaluate microbial growth. **Methods:** HubCentra84 was compared to BacT/Alert^®^ 3D and BACTEC™ FX. *Staphylococcus aureus*, *Streptococcus pneumoniae*, *Escherichia coli*, *Pseudomonas aeruginosa*, *Bacteroides fragilis*, and *Candida albicans* were selected as representative clinically infectious microorganisms. Colonies from pure cultures were diluted with 0.9% saline to create simulated sepsis specimens (SSSs). The SSSs were injected into dedicated culture bottles for each instrument. Thirty paired tests were performed for each strain. **Results:** Colony-forming units of the added SSSs were consistent according to bacteria, and all strains demonstrated robust growth in three CMBCSs. Time-to-positivity was uniformly observed according to the instruments used. The novel CMBCS detected the growth of the clinically significant bacteria *S. aureus*, *S. pneumoniae*, *E. coli*, and *P. aeruginosa* approximately 2 h faster than the other two systems. However, it was approximately 200 min slower for *C. albicans* and 3000 min for *B. fragilis*. **Conclusions:** The novel CMBCS demonstrates advantages in detecting the growth of common clinical bacteria. Although slow growth was detected for certain microorganisms, it successfully captured the growth of all tested microorganisms.

## 1. Introduction

Sepsis is an extremely dangerous and life-threatening disease with mortality rates increasing by 4% every hour of delayed treatment, emphasizing the importance of timely intervention [1,2]. Modern continuous-monitoring blood culture systems (CMBCSs) have provided faster and more reliable diagnostic results to address the limitations of traditional culture methods, which require continuous manual observation after inoculating the culture media. While systems such as BacT/Alert^®^ 3D (bioMérieux, Marcy-l’Étoile, France) and BACTEC™ FX (Becton Dickinson and Company (BD), Franklin Lakes, NJ, USA) have been among the most widely used, these were initially developed in the 1970s [3,4,5].

In the BacT/Alert^®^ system, bacterial growth is detected through pH changes caused by hydrogen ions produced from dissolved carbon dioxide in a culture bottle medium, requiring a colorimetric detector [6]. In contrast, the BACTEC™ FX system also detects hydrogen ions but measures dissolved carbon dioxide through a fluorescence reaction, necessitating separate excitation and emission filters for this fluorescence response [7]. Additionally, some devices employ a completely different method, measuring pressure changes within the bottle, which reflect gasses consumed and produced by microbial metabolism [8]. Although the specific detection mechanisms vary among systems, they all fundamentally confirm carbon dioxide production by microorganisms growing and dividing within the bottle.

While the CMBCS was designed to reduce time and labor significantly, this instrument demanded other essential resources, including a budget for optimal performance, infrastructure for reliable operation, and a suitable environment to contain consumables stably [9]. These requirements posed challenges to the optimal functioning of CMBCSs [10,11]. Consequently, smaller and more affordable systems have been developed and designed to meet the needs of these settings with limited space and financial resources [9]. Although fully automated systems offer convenience, relying solely on such systems means accepting the inevitable challenges associated with achieving faster detection times [4,12]. These challenges often include complex maintenance requirements and other operational issues that come with advanced automation technologies [12,13].

A broad range of microorganisms possess the potential to cause bloodstream infections. However, the World Health Organization (WHO) has prioritized certain infectious syndromes and focused surveillance efforts, via the Global Antimicrobial Resistance Surveillance System (GLASS), on eight core pathogens: *Acinetobacter* spp., *Escherichia coli*, *Klebsiella pneumoniae*, *Salmonella* spp., *Staphylococcus aureus*, *Streptococcus pneumoniae*, *Shigella* spp., and *Neisseria gonorrhoeae* [14]. Among the pathogens commonly identified in bloodstreams, *S. aureus*, *S. pneumoniae*, and *E. coli* are particularly significant, given their prevalence and clinical impact [15,16]. In the Republic of Korea, *P. aeruginosa* accounts for approximately 1.7% of Gram-negative bacterial blood stream infections, a percentage notable due to the pathogen’s multidrug resistance. This has positioned *P. aeruginosa* alongside *S. aureus*, *S. pneumoniae*, and *E. coli* as sepsis pathogens of persistent concern and critical for ongoing monitoring [15,16,17]. In terms of anaerobic bacteria, *Bacteroides fragilis*, an obligate anaerobe, requires a strictly anaerobic environment for cultivation and is commonly grown on blood agar plates [18]. *B. fragilis* was recognized as a prevalent clinical anaerobe associated with high mortality rates in several institutions in the Republic of Korea [19,20]. The WHO has also formally emphasized the need to address bloodstream infections caused by *Candida* species, with the Republic of Korea actively participating in this global initiative [21]. In a study by Won et al., *Candida* spp. were identified as the fifth most common cause of bloodstream infections, with *C. albicans*, *C. tropicalis*, *C. glabrata*, and *C. parapsilosis* frequently observed, especially among individuals over 60 years of age [22].

The identification and characterization of clinical microorganisms in the Republic of Korea are critical issues that have raised a significant amount of interest both among the nation and individuals. While CMBCSs in widespread use are designed for global applicability, a pressing issue has emerged concerning certain systems and media that exhibit notably high false positive rates [12]. In response to growing interest and demand, a domestic CMBCS was developed and tailored to the local environment and resources. This new system drew significant interest regarding its performance compared to existing automated systems, particularly in detecting clinically relevant microorganisms of national interest. This study analyzed whether the newly developed CMBCS demonstrates equivalent or superior time to positivity compared to existing systems for representative clinical strains.

## 2. Materials and Methods

### 2.1. Microorganism Preparation

Our research aimed to evaluate the practical utility of this piece of equipment, HubCentra84 (HUFIT Inc., Chuncheon-si, Republic of Korea), in real-world settings by focusing on microorganisms of particular relevance in the Republic of Korea [16]. This analysis involved a selection of representative pathogens known to cause clinical infections, as well as microorganisms commonly used for quality control in laboratory instruments and consumables. Specifically, the Gram-positive bacteria selected for this study included *S. aureus* and *S. pneumoniae*, which are known to be frequent pathogens in clinical infections. For Gram-negative bacteria, *E. coli* and *P. aeruginosa* were chosen due to their clinical significance in various infections. Additionally, the study incorporated *B. fragilis* as a representative anaerobic bacterium and *C. albicans* as a fungal species to cover a broader spectrum of microbial types that the equipment may encounter in clinical settings. All strains were purchased from American Type Culture Collection and have been continuously managed in a laboratory through pure culture. Once these required microorganisms were identified, fresh pure cultures were prepared on the day prior to testing. For cultivating pure cultures, specific growth media were employed to optimize the growth conditions for each microorganism. For the Gram-positive bacteria, sheep blood agar, the name of which was BAP (BANDIO BioScience Co., Pocheon-si, Republic of Korea), was used, providing a suitable nutrient-rich environment. For the Gram-negative bacteria, including *E. coli* and *P. aeruginosa*, MacConkey agar named MAC (BANDIO BioScience Co.) was used to support selective growth and differentiation. *B. fragilis*, being an anaerobic bacterium, was cultured on BRUCELLA Blood Agar (BANDIO BioScience Co.), supplemented with the GasPak™ EZ Anaerobe Pouch System with an indicator (BD), which was placed with plate agar and enveloped within a pouch to maintain anaerobic conditions. Finally, *C. albicans*, as a fungal representative, was grown on Sabouraud Dextrose Agar with Chloramphenicol (SDAC) (BANDIO BioScience Co.), ensuring suitable conditions for fungal growth.

### 2.2. Instruments with Culture Bottles

HubCentra84 (HUFIT Inc.) was established in a clinical laboratory alongside BACTEC™ FX (BD) and BacT/Alert^®^ 3D (bioMérieux), which are widely used CMBCSs. Each system was provided with a manufacturer-specific aerobic or anaerobic medium. In the BACTEC™ FX (BD), BACTEC™ Plus Aerobic/F Culture Vials (BD) and BACTEC™ Lytic/10 Anaerobic/F Culture Vials (BD) were used as aerobic and anaerobic media, respectively. Similarly, in BacT/Alert^®^ 3D (bioMérieux), BACT/ALERT^®^ SA (bioMérieux) served as aerobic media, while BACT/ALERT^®^ SN (bioMérieux) was designated for an anaerobic culture bottle. Hubix A (HUFIT Inc.) and Hubix N (HUFIT Inc.) were used as the exclusive aerobic and anaerobic culture bottles, respectively, for HubCentra84 (HUFIT Inc.) (Table 1, Appendix A).

### 2.3. Simulated Septic Specimen

In clinical microbiology laboratories, diagnostic specimens are typically prepared by suspending bacterial colonies in a solution calibrated according to the turbidity standards defined by McFarland [23,24]. However, during the early stages of sepsis, the colony-forming unit (CFU) levels in the bloodstream were notably low. In neonates, the CFU levels were often less than 10 CFU/mL, while in adults, they were generally under 100 CFU/mL [25,26]. To mimic these low bacterial concentrations observed in early sepsis, this study involved diluting samples to create simulated septic specimens (SSSs) (Figure 1).

Various studies have reported that inoculating body fluids into culture bottles and observing them in CMBCSs demonstrated a high concordance rate with identification results from conventional culture methods [27,28,29,30]. This transformed approach had enabled clinical physicians to perform pathogen culture testing on a wide range of specimens. As the name implied, a CMBCS is a specialized system designed for diagnosing bloodstream infections, demonstrating superior detection performance with blood specimens [31]. Therefore, this experiment aimed to evaluate the system’s performance in challenging scenarios, such as with non-blood specimens, by preparing SSSs using pure cultured colonies suspended in 0.9% sodium chloride as 0.9% saline [32].

To standardize the bacterial solution concentration, the initial McFarland standard was set to 0.33 using the DENSICHEK^®^ Plus Standards Kit (bioMérieux), providing a baseline for further dilution. In a preliminary test, starting at 0.33 McFarland and proceeding with two dilution steps—1000-fold and 4000-fold—proved to be the most rapid approach. Consequently, the dilution began roughly at 0.33 McFarland before progressing to SSS preparation. On each experimental day, culture bottles were prepared to receive 3 mL of the SSS, with the total volume of the SSS calculated according to the study’s requirements. The total volume of the SSS required for the day of experiment was prepared as a single bulk solution with adequate concentration according to the microorganism, and the time taken to dispense it into all bottles did not exceed 30 min, maintaining sample consistency.

### 2.4. CFU Verification and Growth Confirmation

A 5 mL syringe was used to inject an SSS into each culture bottle. Because of the differing nutrient composition of each medium, procedures were implemented to reduce the carryover contamination bias using syringes.

The entire SSS was divided into several culture bottles requiring inoculation;A single syringe was used to inoculate no more than five bottles;After the inoculation of one type of culture bottle, a new syringe was used;After 3 mL inoculation, 250 µL of the remaining divided sample was used to verify the actual CFU, of which the target range was <10 CFU/mL, by inoculating it onto the solid plate initially used for pure culture.

Culture bottles mounted on the same date were drawn out when a positive growth signal was detected. The bottles were then randomly selected by instrument and media type. Specimens were carefully drawn from the selected culture bottles and inoculated onto the same type of medium originally used for pure culture. This step ensured that any subsequent growth observed in these inoculations would be consistent with the conditions that the initial colonies had experienced, allowing for an accurate comparison of microbial growth and detection across samples. By re-culturing on the same medium, this study minimized variability that could have arisen from differences in media properties. After re-inoculation, the samples were monitored until visible colonies appeared on the medium. Once colony growth was established, each colony was subjected to identification using the MALDI Biotyper Smart System (Bruker Daltonics GmbH, Bremen, Germany). This method ensured consistency in identifying colonies from different bottles and media types, contributing to the comprehensive evaluation of the CMBCS instruments across a range of practical scenarios.

### 2.5. Statistical Analysis

To ensure statistically valid comparisons, 30 paired tests per microorganism were conducted for each system, resulting in 90 paired culture bottle tests per strain. The performance of each system was evaluated based on time-to-positivity detection. Because not all tests were performed on a single day, histograms of all results were gathered, and histograms of time to positivity were generated to verify consistency and normality across the test results. The performance of the systems was compared based on time to positivity. Therefore, the independent variable was a nominal value representing systems, while the dependent variable was a continuous value denoting time to positivity. The Wilcoxon rank-sum test was used to determine statistical significance, with a threshold of *p* < 0.05 set as the criterion for significance, and all analyses were performed using R software 4.4.2 (R Foundation for Statistical Computing, Vienna, Austria).

## 3. Results

The experimental schedule is presented in Appendix A. The dilution factor and CFU concentration of SSSs for each bacterial strain are listed in Appendix A. Although some media for CFU verification showed no colony growth (Appendix A show the actual inoculation images), all culture bottles showed positive growth signals. The confirmation of growth indicated that all strains inoculated on the same day were accurately identified as the intended strains.

*S. aureus*, *S. pneumoniae*, and *E. coli* grew in both aerobic and anaerobic media across all instruments. Consistently, among the three systems, *P. aeruginosa* grew in all aerobic media, whereas *B. fragilis* grew exclusively in anaerobic media. *C. albicans* grew in both aerobic and anaerobic bottles only in HubCentra84 (HUFIT Inc.), whereas it grew solely in aerobic media in the other two systems. For *B. fragilis*, time to positivity showed high reproducibility only in BACTEC™ FX (BD), with the other two instruments displaying distributions with low kurtosis (Figure 2A). *S. aureus* and *E. coli* exhibited faster time-to-positivity values in anaerobic media for all instruments, whereas *S. pneumoniae* exhibited faster time to positivity in aerobic media. *C. albicans* cultured in Hubix N (HUFIT Inc.) displayed faster time to positivity than *C. albicans* cultured in Hubix A (HUFIT Inc.) (Figure 2B).

Blood cultures were performed in aerobic and anaerobic bottles. If either bottle signal is positive, the identification of the infectious pathogen can proceed more rapidly. For each strain, only the earliest time to positivity from each pair was selected for comparison (Figure 3). The strains representing Gram-positive and Gram-negative bacteria, *S. aureus*, *S. pneumoniae*, *E. coli*, and *P. aeruginosa*, demonstrated time-to-positivity values that were approximately 2 h faster in HubCentra84 (HUFIT Inc.) than in the other systems. Although *C. albicans* grew in both media types only in HubCentra84 (HUFIT Inc.), its time to positivity was 200–300 min slower than in the other two instruments. Notably, *B. fragilis* exhibited a delay of approximately 3000 min.

## 4. Discussion

Microorganisms growing within culture bottles produced carbon dioxide, which consequently altered the medium environment. The principle of detecting these changes through engineering methods was applied in CMBCSs, with specific mechanisms differing by instrument. Typically, these systems measured dissolved carbon dioxide levels in the medium either through colorimetry or fluorescence reactions. Colorimetric methods were particularly integrated into newer instruments. Hardy et al. conducted a comparative analysis of four emerging CMBCS instruments manufactured in China, all of which utilized colorimetry [9]. The HubCentra84 system also operated on a colorimetric basis, contributing to component simplification within the device (Table 1). Nolte et al. analyzed the performance of an early BACTEC model, noting that the system incorporated excitation and emission filters to detect fluorescence reactions [7]. With the colorimetry, the detection component was positioned outside the culture bottle, enabling color changes to be observed through the outer material of the bottle itself. Chen et al. reported that technological advancements capable of addressing the enzyme-like activity of nanomaterials refined measurement techniques using colorimetry and fluorescence analysis for bioactive substances [33]. Among these, colorimetry, which required simpler hardware, achieved significant progress in diagnosing infectious diseases and antimicrobial resistance through a broader range of methods [34]. While this design could have been perceived as a potential obstacle, the simplified setup achieved a comparable or even enhanced performance relative to earlier systems, suggesting that it serves as a breakthrough in CMBCS technology.

The HubCentra84 (HUFIT Inc.) demonstrated considerable strength in detecting the growth of *S. aureus*, *S. pneumoniae*, *E. coli*, and *P. aeruginosa*, outperforming two alternative devices by reducing the detection time by up to two hours (Figure 2). However, *C. albicans* and *B. fragilis* exhibited a more delayed time to positivity compared to the two existing systems. Our research team hypothesized that this discrepancy was due to differences in media composition. *C. albicans* was typically recommended to be cultured in aerobic media [35]. This result aligned with our study, where *C. albicans* grew 100% only in the aerobic media of the two existing systems (Figure 2). Similar to how *S. aureus* inhibited growth in the presence of 1.0 M sodium chloride [36], *C. albicans* was affected by sodium ions at pH 8.0, which suppressed germ tube formation, while chloride ions further accelerated this suppression [37]. For this reason, these factors likely caused the delayed growth of *C. albicans* in aerobic media.

Interestingly, *C. albicans* grew in the anaerobic media of HubCentra84 (HUFIT Inc.) but not in the anaerobic media of the other two systems. L-cysteine was known to undergo N-acetylation to form N-acetylcysteine (NAC), which has long been recognized for its mycolytic effects [38]. Phuengmaung et al. reported that NAC attenuated biofilm thickening in *C. albicans*, thereby inhibiting its proliferation [39]. However, several studies have consistently reported that NAC does not affect the growth of *C. albicans*, with recent findings revealing that high concentrations of NAC inhibit growth, while low concentrations promote it [40,41]. As shown in Table 1, the anaerobic media of HubCentra84 (HUFIT Inc.) contained L-cysteine. Although the exact concentration was unknown, it was hypothesized that a low concentration of L-cysteine in the anaerobic media might have facilitated the growth of *C. albicans*, resulting in faster time to positivity compared to the aerobic media.

Although *Candida* spp. generally exhibit robust growth in aerobic CMBCS bottles [5,42], the HubCentra84 (HUFIT Inc.) system successfully detected *C. albicans* growth in both aerobic and anaerobic bottles (Figure 2A), albeit with a four-hour delay compared to other systems (Figure 2B). The detection of *C. albicans* in anaerobic media suggests an extended environmental adaptability, allowing for the potential growth of other *Candida* species in CMBCSs, which could ultimately enhance pathogen identification in clinical laboratories. However, the delayed detection time, when compared to contemporary devices, underscores a limitation that warrants further attention. An increase in the detection efficiency of fungal pathogens, particularly in anaerobic settings, could play a critical role in improving patient outcomes by enabling earlier intervention.

A study by Menchinelli et al. demonstrated that *B. fragilis* could be effectively cultured in CMBCS bottles containing donated human blood, with typical growth detected after 30 h [4]. In this study, we used SSSs prepared with 0.9% saline rather than blood to conduct a comparative experiment among CMBCS instruments. Both the BACTEC™ FX (BD) and BacT/Alert^®^ 3D (bioMérieux) systems generated time-to-positivity values within approximately 2000 min. In contrast, HubCentra84 (HUFIT Inc.) required around 5000 min to produce time to positivity (Figure 3). To explain the significant difference observed, we focused on the sodium thioglycolate present in the anaerobic media of HubCentra84 (HUFIT Inc.) (Table 1). Thiol and Brewer’s thioglycolate media have long been recognized as effective for cultivating *B. fragilis* [43]. However, Tamimi et al. previously reported that extremely low or high concentrations of sodium thioglycolate inhibited the growth of *B. fragilis* [44]. The fact that *B. fragilis* grew well only within a specific concentration range posed a significant challenge when using it as an additive in culture bottles. This was because the volume of blood collected varied greatly depending on the operator’s proficiency and the patient’s condition. Despite the confirmed rapid time to positivity for major clinical pathogens, the extended detection time for *B. fragilis* with HubCentra84 (HUFIT Inc.) presents a significant challenge.

Conducting blood culture tests in a clinical microbiology laboratory carried significant implications. It required various essential equipment, including instruments to culture, devices to identify, and systems to perform antimicrobial susceptibility testing on pathogens [45]. Leo et al. stated that with the commercialization of machine learning and next-generation sequencing, full automation in clinical microbiology laboratories has become possible; however, they emphasized once again the necessity of having fundamental equipment in place [45]. As shown in Table 1, HubCentra 84 (HUFIT Inc.) had half the instrument height of comparable devices, highlighting its compact, space-saving design. Furthermore, as observed in Appendix A, the modular configuration of the instrument allowed for efficient stacking and usage. Additionally, HubCentra 84 (HUFIT Inc.) was significantly lighter than other instruments, reducing the risk of damage during transport and installation. For newly established clinical microbiology laboratories initiating blood culture testing, the novel instrument represented an attractive option.

This study has some limitations. First, since CMBCSs were developed specifically for blood culture, their performance was optimal when the specimen was blood. The use of 0.9% saline instead of blood to prepare SSSs was a significant limitation. Kim et al. reported that *B. fragilis* was particularly well identified in blood specimens [46]. However, in actual clinical settings, the identification of pathogens in body fluids was also commonly performed using CMBCSs [27,28,29,30]. In this study, a more challenging scenario was simulated to comparatively analyze the systems’ performance. Nonetheless, future studies would need to incorporate artificial septic blood samples using actual blood specimens and evaluate the systems’ limit of detection for a more comprehensive assessment. Second, a variety of bacteria were not compared. In the modern era, with an aging population and a large number of therapeutically or pathogenically immunosuppressed patients, clinical laboratories have encountered a completely different spectrum of pathogens. Although six representative microorganisms were included, evaluating the utility of CMBCSs in contemporary clinical laboratories would have required testing a greater number of pathogens. Third, uniform CFU concentrations across the various samples were not tested. Although efforts were made to compare consistent CFU levels among CMBCSs, identical CFU counts across bacterial strains were not achieved. This limits the accuracy of the performance comparison for each strain. The detection limit of the system was determined by calibrating the dilution factor using a wide range of CFU concentrations. Fourth, actual blood samples from patients with sepsis were not used. Clinical blood specimens offer an ideal means to compare CMBCS performance. However, collecting an additional 20 mL of blood from patients with septic shock for experimental purposes raises ethical concerns. By further improving the performance of the CMBCS and confronting real-world assessments in clinical fields, a rigorous performance evaluation by end users could be achieved.

## 5. Conclusions

Our study compared the performance of existing CMBCSs with that of a newly developed automated blood culture system. Efforts were made to ensure the coherence, consistency, and plausibility of the comparative results. The findings revealed that the new CMBCS demonstrated earlier growth detection for clinically common bacteria, such as *S. aureus*, *S. pneumoniae*, *E. coli*, and *P. aeruginosa*, compared with other systems. However, further performance improvements are deemed necessary for *C. albicans*, a pathogen of recent concern, and *B. fragilis*, which is frequently encountered in clinical laboratories. The emergence of novel systems, the advancement of stagnant performances, and the recognition of limitations are all expected to contribute significantly to improvements in clinical microbiology.

## Figures and Tables

**Figure 1 diagnostics-15-00468-f001:**
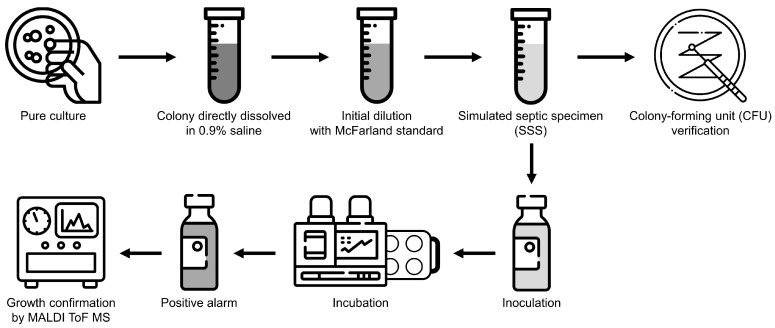
A schematic flow of the experiment. On the day of the experiment, the required SSS for each selected clinical representative microorganism is prepared in bulk, then aliquoted for each system to eliminate carryover contamination during inoculation. The remaining SSS after inoculation undergoes verification to ensure that the same CFU level of the pathogen is inoculated in each automated instrument. Additionally, bottles triggering a positive signal are subjected to identification of the growth microorganism to confirm consistency with the initially inoculated pathogen by matrix-assisted laser desorption/ionization time-of-flight mass spectrometry (MALDI ToF MS).

**Figure 2 diagnostics-15-00468-f002:**
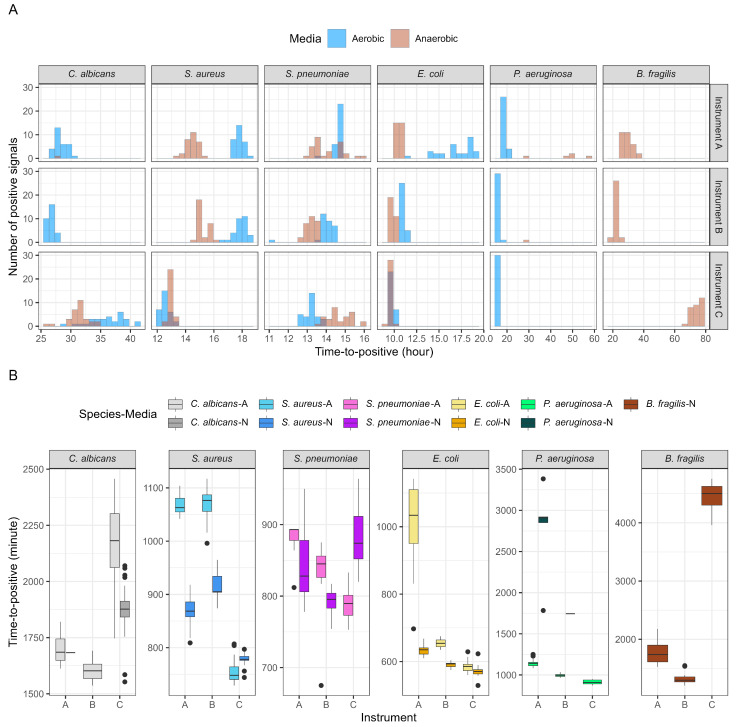
Time to positivity of continuous monitoring bacterial culture systems (CMBCSs). (**A**) A histogram illustrating the time-to-positivity values reported by each CMBCS after inoculation with simulated septic specimens prepared using clinically representative infection strains. The *x* axis denotes time to positivity, which is divided into 12 h intervals. The height of the column represents the number of positive signals within each interval. The column color indicates the media type. As the columns were semi-transparent, overlapping columns for the two media types were visible. (**B**) The box represents the interquartile range (IQR) of time-to-positivity values, and the bold horizontal line in the center of the box indicates the median. Whisker lines denote values extending up to 1.5xIQR from each boundary of the box. Black dots indicate outliers that fall beyond the whisker range. The box color specifies the inoculated strain and media type. A number of illustrated boxes mark the number of media types in which each strain grows. Abbreviations: instrument A, BacT/Alert^®^ 3D (bioMérieux, Marcy-l’Étoile, France); instrument B, BACTEC™ FX (Becton Dickinson and Company (BD), Franklin Lakes, NJ, USA); instrument C, HubCentra84 (HUFIT Inc., Chuncheon-si, Republic of Korea); media A, aerobic; media N, anaerobic.

**Figure 3 diagnostics-15-00468-f003:**
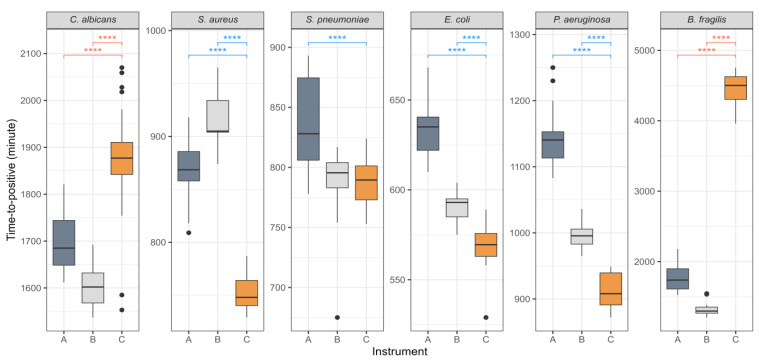
The rapidity of time to positivity by instrument. The connecting brackets represent the instruments compared. The Wilcoxon rank-sum test is used to analyze the statistical significance of differences between instruments for time to positivity. Asterisks in brackets indicate *p* for time to positivity. All *p* values are <0.001; therefore, the values are the same in ****. The color of the statistical results varies according to the outcomes of instrument C, and the color of the box denotes the instrument. Abbreviations: instrument A, BacT/Alert^®^ 3D (bioMérieux, Marcy-l’Étoile, France); instrument B, BACTEC™ FX (Becton Dickinson and Company (BD), Franklin Lakes, NJ, USA); instrument C, HubCentra84 (HUFIT Inc., Chuncheon-si, Republic of Korea).

**Table 1 diagnostics-15-00468-t001:** Details of compared systems.

Details	BacT/Alert^®^ 3D ^1^	BACTEC™ FX ^2^	HubCentra84
Instrument			
Capacity (cells)	240	200	84
Dimensions (width × depth × height, mm)	838 × 608 × 900	635 × 864 × 1016	650 × 552 × 490
Weight (kg)	174	251	47
Agitation	Continuous rocking	Continuous rocking	Continuous rocking
Detection	Colorimetry	Fluorescence	Colorimetry
Control system	LCD touch pad	LCD touch pad and EpiCenter™ running on Windows OS	Control program on a laptop running Windows OS
Media ^3^			
Aerobic	Pancreatic digest of casein, papaic digest bean meal, SPS, pyridoxine HCl, other complex amino acid and carbohydrate substrates in purified water	Soybean–casein digest broth, yeast extract, dextrose, sucrose, hemin, menadione, pyridoxal HCl, SPS, nonionic absorbing resin, cationic excahge resin, processed water	Casein, soybean meal, dextrose, dibasic potassium phosphate, sodium chloride, polyanetholesulfonic acid, resins
Anaerobic	Pancreatic digest of casein, papaic digest bean meal, SPS, menadione, hemin, reducing agents, other complex amino acid and carbohydrate substrates in purified water	Soybean–casein digest broth, yeast extract, animal tissue digest, dextrose, hemin, menadione, sodium citrate, thiols, sodium pyruvate, saponin, antifoaming agent, SPS, processed water	Casein, yeast extract, dextrose, dibasic potassium phosphate, sodium chloride, L-cystine, sodium thioglycolate, polyanetholesulfonic acid, resins

^1^ Details of combining a control module and an incubator module from the BacT/Alert^®^ 3D 240 (bioMérieux) system. ^2^ Specifications for a single BACTEC™ FX (Becton Dickinson and Company) system on its stand. ^3^ The exact compositional ratio is not known as it is based on information provided by the supplier. Abbreviations: LCD, liquid crystal display; OS, operating system; SPS, sodium polyanetholesulfonate; HCl, hydrogen chloride.

## Data Availability

The data analyzed in this research are available upon request from the corresponding author.

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
