# Peer review of "Comparative Performance Evaluation of Continuous Monitoring Blood Culture Systems Using Simulated Septic Specimen"

_diagnostics, 2025, doi:10.3390/diagnostics15040468_

Round 1

Reviewer 1 Report (Previous Reviewer 1)

Comments and Suggestions for Authors

Recently I had reviewed same manuscript under the submission #3319454.  For this reason I have said Plagiarism.  

If this is a new submission of previous submission ID #3319454 then I will have to admit that there is good amount of improvement.  Still some more clarity is required, which I have written directly in the MS as highlights or comments, which author may want to refer to.

Author Response

Comment 1. Recently I had reviewed same manuscript under the submission #3319454.  For this reason I have said Plagiarism. If this is a new submission of previous submission ID #3319454 then I will have to admit that there is good amount of improvement. Still some more clarity is required, which I have written directly in the MS as highlights or comments, which author may want to refer to.

Response 1. We sincerely appreciate your keen attention to detail and your effort in recalling the contents of the previous manuscript with such clarity, as well as your vigilance regarding potential copyright issues. The manuscript you have reviewed this time is a resubmission incorporating the experimental improvements suggested by the editor for manuscript #3319454. The authors have thoroughly reviewed the PDF containing all of your comments and have prepared an answer note accordingly. We are truly grateful for your dedication to enhancing the quality of the resubmitted manuscript, and we deeply appreciate your valuable insights and constructive feedback.

Comment 2. physiological saline. (line 23)

Response 2. We deeply appreciate your insightful comments, especially for highlighting a crucial terminology that could have been overlooked. In our institution, a 0.9% sodium chloride solution had commonly been referred to as “normal saline” for simplicity. In this study, we used this “normal saline” to simulate infected human specimens, as you had correctly pointed out. However, compared to real physiological saline, 0.9% sodium chloride saline indeed lacks many electrolytes. Therefore, to convey a more precise meaning when referring to the solution used in preparing simulated septic specimen (SSS) in this experiment, the term "0.9% saline" was explicitly used.

Comment 3. trace of oxygen? (line 29)

Response 3. One of the key findings of this study was that Bacteroides fragilis exhibited significantly slower growth in the novel instrument compared to BACTEC FX and BacT/Alert 3D. If, as the reviewer suggested, a trace amount of oxygen had been present in the culture bottle, B. fragilis, which was an obligate anaerobe, would not have been able to grow. Although delayed, the novel instrument's culture bottle eventually generated a clear growth signal for B. fragilis. Furthermore, subculturing the sample from the same culture bottle onto solid agar resulted in colonies that were definitively identified as B. fragilis through MALDI ToF MS identification. Based on these findings, it was hypothesized that the composition of the culture bottle media played a role in the delayed growth, with sodium thioglycolate being the key factor. A review of previous studies revealed that sodium thioglycolate has an optimal concentration range for supporting the growth of B. fragilis. These findings are discussed in detail in the fifth paragraph of “Discussion” section. However, due to space constraints, the abstract only provides a concise statement of the key facts rather than a full explanation. Furthermore, since comparative experiments with varying concentrations of the suspected substance were not conducted, mentioning this in the abstract could potentially cause confusion among readers.

Comment 4. Why 0.33 McFarland Standard? (figure 1)

Response 4. During this experiment, the primary focus was to prepare a simulated septic sample (SSS) by suspending colonies from solid agar in physiological saline before inoculation. To ensure that all prepared culture bottles were inoculated within 30 minutes. To maintain this speed, the number of dilution steps was minimized. In a preliminary test, starting at 0.33 McFarland and proceeding with two dilution steps—1,000-fold and 4,000-fold—proved to be the most rapid approach. Consequently, the dilution began roughly at 0.33 McFarland before progressing to SSS preparation. This information has been incorporated into the main text, while the figure was revised to omit the 0.33 value.

Comment 5. Physiological saline and not a normal saline. (figure 1)

Response 5. I appreciate the attention given to both the figure and the main text. As mentioned in the previous comment, the dilution solution used in this experiment is neither "normal" nor "physiological". Therefore, it was explicitly referred to as "0.9% saline". A relevant reference was cited, and the term "saline" was revised accordingly in both the figure and the subsequent manuscript.

Comment 6. Is the growth confirmation is on the basis of OD or Cfu/ml? (figure 1)

Response 6. I agree that the phrase "growth confirmation" alone may have hindered clear communication. The exact method has been specified in Section 2.4. CFU Verification and Growth Confirmation. In the CMBCS, when a positive alarm was triggered, a small amount of the sample from the culture bottle was inoculated onto solid agar to form colonies. These colonies were then identified using Matrix-Assisted Laser Desorption/Ionization Time-of-Flight Mass Spectrometry (MALDI ToF MS) to confirm whether the microorganism was the same as the one originally introduced into the culture bottle. Figure 1 has been revised to indicate that growth confirmation was performed using MALDI ToF MS.

Comment 7. How much of volume was inoculated is not clear. Was it 1 ml per bottle or whole 5 ml per bottle?  What was the medium volume in Bottle? Why can't inoculum be expressed in terms of colony forming units per bottle or per ml? (line 199)

Response 7. I am grateful for the feedback regarding the missing information in the summary. Steps 1 to 4 were intended to summarize the content of Sections 2.3 and 2.4. In an effort to condense the information, some essential details were inadvertently omitted. The inoculated volume per bottle was 3 mL. The exact volume of the medium in each bottle was not specified in the manufacturer's insert, making it impossible to determine precisely. The inoculum was adjusted to ensure that the final concentration remained below 10 CFU. These details have now been included.

Comment 8. Why is that the inoculum of Bacteroides fragilis (ATCC 25285) strain was nearly 2 to 2.5 higher than other inocula? Why was Escherichia coli and Pseudomonas aeruginosa at same dilution factor gave less than 40% of common inoculum? Would this differential concentration affect growth speed? Despite providing such a high inoculum concentration growth of B. fragilis was albeit slow? Plausibilities need to to be proposed in the light of published literature.

Response 8. The questions raised by the reviewer are identical to the concerns previously highlighted by the editor in a prior manuscript, where a re-experimentation was suggested. In the pre-revision study, Staphylococcus aureus, Streptococcus pneumoniae, and Bacteroides fragilis were inoculated at 1,377 CFU/mL, 172 CFU/mL, and 254 CFU/mL, respectively. Consequently, the editor recommended conducting the experiment at an adjusted bacterial concentration of 10 CFU/mL, which reflects the early-stage sepsis blood concentration as initially proposed by the authors. In this revision, the inoculation was adjusted to match the dilution factor applied to Escherichia coli and Pseudomonas aeruginosa, which had the lowest concentrations in the previous experiment. One possible explanation for the observed discrepancy is the use of different solid agar types for CFU verification. However, ensuring that the final inoculum concentration remained below 10 CFU/mL was considered more significant.

Notably, time-to-positive values of the three bacterial species, whose inoculation concentrations were adjusted and re-experimented, showed no significant differences compared to the previous results. This suggests that each bacterial species has a distinct intrinsic growth rate. The observed variations are likely attributable to differences in the nutrient compositions of culture bottles from different manufacturers. These aspects have been thoroughly addressed in the fifth paragraph of the Discussion section, which was newly emphasized during the previous revision. The reviewer’s comment aligns with this revision.

Reviewer 2 Report (Previous Reviewer 2)

Comments and Suggestions for Authors

The authors compare a new continuous monitoring blood culture system (CMBCS) HubCentra84 with two established systems for detecting microbial growth in simulated sepsis specimens. The new system showed faster detection for common clinical bacteria but was slower for Candida albicans and Bacteroides fragilis. Overall, HubCentra84 successfully detected growth for all tested microorganisms demonstrating its potential as an alternative CMBCS.

The study is interesting but I have a few comments that need to taken into consideration:

1. Line 55: The statement "While CMBCS was designed.." could be phrased clearly to avoid confusion between CMBCS and automated instruments.

2. The introduction mentions South Korea and the Republic of Korea interchangeably. It's better to use one consistent term.

3. Line 98: "Our research aimed.." is vague. It can be made more specific by referring to the equipment.

4. Line 116: The explanation of MacConkey agar is unnecessarily detailed. It can be removed. 

5. Line 127- The sentence is confusing and can be rewritten for clarity. The phrase "real set in clinical laboratories" is unclear and can be rephrased.

6. Line 151- "These standards.." can be removed.

7. Methods contain too much information which is not necessary. The method should be specific and explain why and how the experiment was conducted rather than pointing out the definitions.

8. Table 2 can be moved to supplementary.

9. The figure legends should include the statistical test used for significance wherever needed.

Author Response

Comment 1. Line 55: The statement "While CMBCS was designed.." could be phrased clearly to avoid confusion between CMBCS and automated instruments.

Response 1. I appreciate the precise identification of the ambiguous section. The terms CMBCS and automated instrument referred to the same system in the sentence in question. To clarify this, "automated instrument" was replaced with "this instrument" to explicitly indicate that both terms refer to the same entity.

Comment 2. The introduction mentions South Korea and the Republic of Korea interchangeably. It's better to use one consistent term.

Response 2. I am grateful for the careful review in identifying potentially confusing sections. All instances were revised to "Republic of Korea", and this change was consistently applied throughout the entire manuscript.

Comment 3. Line 98: "Our research aimed.." is vague. It can be made more specific by referring to the equipment.

Response 3. I acknowledge the precise identification of the missing information. It is true that "this equipment" lacked a clear reference, which could have led to further confusion. To clarify, the mention of the equipment was moved forward in the sentence for explicit identification.

Comment 4. Line 116: The explanation of MacConkey agar is unnecessarily detailed. It can be removed.

Response 4. The feedback on excessive explanation making readability difficult was duly noted. The description of MacConkey agar was removed, and the corresponding reference was also deleted. Consequently, the numbering of the references was reorganized throughout the manuscript.

Comment 5. Line 127- The sentence is confusing and can be rewritten for clarity. The phrase "real set in clinical laboratories" is unclear and can be rephrased.

Response 5. The numerous comments provided to aid understanding from both a reader’s and an expert’s perspective were highly valued. The highlighted sentence, which simply described the comparison of three instruments, was unnecessarily complex. It was revised into a more concise single sentence.

Comment 6. Line 151- "These standards.." can be removed.

Response 6. I greatly admire the consistent efforts in identifying ambiguities. The sentence in question was removed.

Comment 7. Methods contain too much information which is not necessary. The method should be specific and explain why and how the experiment was conducted rather than pointing out the definitions.

Response 7. The excessive complexity in the Method section was acknowledged, and thanks were given to the reviewer for the considerate revision suggestions. The detailed explanation of the systematic experimental design aimed at minimizing errors was overly elaborate. To improve clarity, non-essential subjective information was eliminated, and the section was condensed accordingly.

Comment 8. Table 2 can be moved to supplementary.

Response 8. Table 2 was considered an experimental reference and was therefore moved to the supplementary table.

Comment 9. The figure legends should include the statistical test used for significance wherever needed.

Response 9. The statistical methods used were described in the figure that includes the results of the statistical analysis.

This manuscript is a resubmission of an earlier submission. The following is a list of the peer review reports and author responses from that submission.

Round 1

Reviewer 1 Report

Comments and Suggestions for Authors

Authors have chosen a good topic and much needed to expedite diagnosis as well as accuracy of diagnosis.  

A few comments for authors consideration. 

1. Acronyms be not made capital, e. g. Escherichia coli -preferred Eco not ECO, which may be adopted to all organisms.

2. Authors need to improve discussion providing sound reasoning on delay imparted in growth of Candida albicans and Bacteroides fragilis.  This might ensure better utility of proposition made here in the manuscript.

3. workflow provided does not indicate growth at terminal point is confirmed only by OD or CFU/ml

Author Response

Comments 1: Acronyms be not made capital, e. g. Escherichia coli -preferred Eco not ECO, which may be adopted to all organisms.

Response 1: I sincerely appreciate your insightful observation regarding the potential for confusion in the notation of microorganisms used in this study. I fully concur with your point that "ECO" does not intuitively indicate a bacterial species. However, "Eco" could similarly lead to ambiguity, as it is a common noun with other meanings. To address this, we have revised all acronyms throughout the manuscript and figures to reflect the original nomenclature of the bacterial species. We trust that this change will effectively clarify the content for readers.

Comments 2: Authors need to improve discussion providing sound reasoning on delay imparted in growth of Candida albicans and Bacteroides fragilis. This might ensure better utility of proposition made here in the manuscript.

Response 2: I appreciate the recognition of this as the most interpretive aspect of the study. Among the microorganisms studied, C. albicans and B. fragilis exhibited different patterns compared to the other four strains. Our research team attributed these differences to variations in culture media composition, consulted the manufacturers regarding the media formulation, and included a discussion on the differences between the three systems.

  1. albicans is inhibited in germ tube formation by sodium chloride but has its biofilm formation promoted by low concentrations of N-acetylcysteine. This explains its delayed growth in the aerobic media of the two other systems, which contain sodium chloride, and its 100% growth in the anaerobic media of HubCentra84, likely due to the presence of L-cysteine in its anaerobic media.
  2. fragilis requires thioglycolate for growth, but only within a specific concentration range. Although the exact concentration was unknown, the extremely delayed time-to-positive observed for B. fragilis in HubCentra84 suggests that the concentration of thioglycolate in its media was inhibitory.

These detailed findings were added to the discussion section to provide a more comprehensive explanation of the results presented in this study.

Comments 3: workflow provided does not indicate growth at terminal point is confirmed only by OD or CFU/ml

Response 3: We deeply appreciate your critical observation regarding the discrepancies in Figure 1, which outlines the methods. As noted, when creating simulated septic specimens (SSS), the concentration was indeed adjusted using the McFarland standard. However, the previous scheme had oversimplified and compressed the process, leading to the final representation of SSS. In the revised Figure 1, we have illustrated the detailed steps involved in creating SSS as described in the methods. Initially, bacterial colonies are suspended in normal saline, resulting in a highly concentrated solution, which is depicted using the darkest grey-grade test tube. This is then diluted to McFarland standard 0.33, represented by a medium grey-grade test tube. Finally, the SSS, further diluted according to the dilution factors detailed in Table 2, is shown as the lightest grey-grade test tube.

Reviewer 2 Report

Comments and Suggestions for Authors

This study evaluates the performance of HubCentra84 against two established systems in detecting microbial growth using sepsis specimens. The study is interesting; however, I have some comments that needs to be addressed:

1. The research question and hypothesis are not well stated in the introduction section.

2. Line 65- The mention of South Korea's participation in WHO's GLASS program if relevant, should be more clearly explained.

3. Line 80- Specify the exact strains or isolates of microorganisms used, including their source.

4. Line 130- The percentage of normal saline used?

5. Line 150- What is this dilution media? What is the final concentration of SSS used?

6. Lien 186- Specify the tests used for significance.

7. In Table 2, the abbreviations of the microorganisms should be written to make it easy to understand the result.

8. Figure 2- What is count in Figure A? 

9. In Figure 3 legend -mention the test used for determining the statistical significance.

10. The first paragraph of the discussion section contains methodological details that would be more appropriate in the Methods section.

11. The discussion could benefit from a more explicit comparison of HubCentra84's overall performance to the existing literature on CMBCS technologies.

Author Response

Comments 1: The research question and hypothesis are not well stated in the introduction section.

Response 1: The sharp and impactful feedback regarding the lack of clarity in the presentation of the research question and hypothesis has been truly lightening. As highlighted in comment 2, the final paragraph of the introduction, which includes content related to WHO's GLASS, was indeed unnecessarily complex and contributed to the lack of precision in articulating the research focus. To address this, the paragraph has been modified following the third paragraph of the introduction. This new section explicitly states that our research team had identified the relatively high false-positive rates in existing CMBCS, and this study aimed to analyze the performance and potential advantages of the newly developed domestic CMBCS.

Comments 2: Line 65- The mention of South Korea's participation in WHO's GLASS program if relevant, should be more clearly explained.

Response 2: We sincerely acknowledge your important critique regarding the sudden introduction of WHO’s GLASS in the introduction section. Your insight further emphasized that the subsequent presentation of the research question and hypothesis (as highlighted in Comment 1) adds to the complexity of the introduction, making it appear disjointed. The mention of GLASS was initially intended to provide justification for selecting six representative clinical pathogens in this study. Currently, the rationale for these six pathogens is discussed in the third paragraph of the discussion, where we detail the surveillance strains from WHO’s GLASS and the AMR monitoring strains from the Kor-GLASS framework. Additionally, the section in the discussion explaining the significance of Candida albicans and Bacteroides fragilis in the context of South Korea has also been relocated to the introduction. All these contents have been structured as a separate paragraph in the introduction. After considering your feedback, we agree that moving this content to the introduction would enhance the logical progression of the manuscript and ensure a clearer narrative for the readers.

Comments 3: Line 80- Specify the exact strains or isolates of microorganisms used, including their source.

Response 3: The detailed observation regarding the omission of the content from Table 2 is deeply appreciated. The strains used as representative clinical strains in this study were not clinical isolates but were instead purchased from the American Type Culture Collection (ATCC). These strains were routinely used for quality checks of laboratory instruments and consumables.

Comments 4: Line 130- The percentage of normal saline used?

Response 4: We greatly appreciate your detailed comment, which even the authors had missed. In the preparation of the SSS, 0.9% normal saline, with a composition similar to body fluids, was utilized.

Comments 5: Line 150- What is this dilution media? What is the final concentration of SSS used?

Response 5: Your point highlights the complexity of the sentence, and we acknowledge the need for greater clarity. The intended meaning is that the total volume of SSS required for the experiment was prepared in a single batch on the day and promptly dispensed into the culture bottles. The term "dilution media" refers to 0.9% normal saline, and "final concentration of SSS" indicates that the entire bulk SSS was adjusted according to the dilution factors outlined in Table 2. As the authors, we express our deep respect for your precise understanding and valuable feedback. The modified sentence was as follow, “The total volume of SSS required for the day of experiment was prepared as a single bulk solution with adequate concentration according to microorganism, and the time taken to dispense it into all bottles did not exceed 30 minutes, maintaining sample consistency.”

Comments 6: Lien 186- Specify the tests used for significance.

Response 6: It is admirable of you to notice even the statistical methods that ensure the validity of the study, which we had completely overlooked. In this research, the statistical analysis employed was the Wilcoxon rank-sum test, with systems designated as independent variables and time-to-positive as the dependent variable. The modified sentence was as follow, “The performance of the systems was compared based on time-to-positive. Therefore, the independent variable was a nominal value representing systems, while the dependent variable was a continuous value denoting time-to-positives. The Wilcoxon rank-sum test was used to determine statistical significance, with a threshold of p < 0.05 set as the criterion for sig-nificance, and all analyses were performed using R software (R Foundation for Statistical Computing, Vienna, Austria).”

Comments 7: In Table 2, the abbreviations of the microorganisms should be written to make it easy to understand the result.

Response 7: Your feedback on maintaining overall consistency throughout the manuscript has been incredibly helpful. In this paper, we initially used acronyms for the various bacterial species mentioned. However, acronyms such as ECO and CAL could potentially evoke other meanings. Therefore, we decided to use the full names of all bacterial species consistently throughout the manuscript. We deeply appreciate your thorough review of our paper with such attention to detail, as if it were your own research.

Comments 8: Figure 2- What is count in Figure A?

Response 8: We greatly value your insightful feedback regarding areas where the meaning was not conveyed clearly. Figure 2A illustrates the time-to-positive for each microorganism as a histogram, with data divided into 12-hour intervals. The "counted" values represent the number of positive signals within each interval. To improve clarity, we have updated the y-axis title of Figure 2A to "Number of positive signals" and added an explanation of the 12-hour intervals to the legend.

Comments 9: In Figure 3 legend -mention the test used for determining the statistical significance.

Response 9: Your effort to ensure consistency between the text and figures, along with your sixth comment, was greatly insightful. We addressed this by stating simply, "Statistical significance is confirmed by Wilcoxon rank-sum test."

Comments 10: The first paragraph of the discussion section contains methodological details that would be more appropriate in the Methods section.

Response 10: Your attention to the logical flow of the entire manuscript is truly meticulous, leaving me deeply appreciative. The first paragraph of the discussion explains the methodological validity of the study, and I agree that, as you pointed out, moving it to the methods section would create a cleaner structure for the paper. Accordingly, this section has been relocated to follow Section 2.4 in the methods.

Comments 11: The discussion could benefit from a more explicit comparison of HubCentra84's overall performance to the existing literature on CMBCS technologies.

Response 11: I appreciate your in-depth understanding of the equipment. HubCentra84 employed a colorimetry-based method, similar to the BacT/Alert 3D system. The first paragraph of the discussion in the previous manuscript focused solely on the differences between the two technologies, colorimetry and fluorescence technology. I agreed with the reviewer's comment that adding a more detailed technical comparison would have enriched the content. Indeed, advancements in nanomaterial control technology had significantly improved the application of colorimetry and fluorescence analysis for bioactive measurements. In particular, colorimetry, which required simpler mechanical equipment, became a foundational diagnostic tool in many fields of clinical microbiology. This additional context was incorporated into the first paragraph of the discussion.

Reviewer 3 Report

Comments and Suggestions for Authors

Major comments:

There is a fundamental problem with this paper in that it does not allow other researchers the opportunity to reproduce the results for themselves. For a commercially available instrument, researchers can purchase or loan the instrument and reproduce the protocol outlined in the methods. However, there is no indication that the HubCentra 84 is commercially available or about to become commercially available. There is no possibility of a researcher reproducing the instrument themselves on the basis of the information provided. If the instrument is commercialized or very close to being commercially available then this should be explained in the paper. Otherwise the data contained in the paper will be of very limited interest to readers of this journal.

I find it strange that the authors did not choose to use sterile blood for preparation of the inocula of microorganisms. If you are going to make “simulated sepsis specimens”, why not include blood? This would have been relatively easy to do. The BactAlert and Bactec bottles are designed to work with blood. The speed of detection might be improved by the presence of blood. Furthermore, it is known that blood cells will release carbon dioxide in blood culture bottles and blood samples with a high white blood cell count can give false positive results. From the results of this paper, it is not known whether the presence of sterile blood would have caused false positive reactions in the HubCentra 84. At the very least, this should be highlighted as a major limitation.

Minor comments

Line 95: How does a desiccant (which is a drying agent) help to maintain anaerobic conditions?

Line 96: Please provide the full name for SDAC medium

Line 97: The challenge set of microorganisms consisted of only 7 strains. Given that 100 or more species may be isolated in any given year from routine blood cultures it is misleading to call this selection “comprehensive”. It would have been useful to include fastidious Gram-negative bacteria such as Haemophilus influenzae or Neisseria meningitidis or Brucella species.

Line 121: “suspension” would be more accurate than “solution”.

Line 166: “Culture bottles containing two or more CMBCS instruments…..”. This does not make sense to me.

MacConkey agar is a selective medium containing bile salts and is not an optimal medium for performing colony counts of Pseudomonas. Counts of Pseudomonas are typically higher on blood agar than on MacConkey agar. There is no reason to use a selective medium when assessing the count of a pure culture.

Line 186: Please detail the statistical tests actually used.

Comments on the Quality of English Language

I would recommend the paper is edited by an expert in English to improve readability.

Author Response

Major comments 1: There is a fundamental problem with this paper in that it does not allow other researchers the opportunity to reproduce the results for themselves. For a commercially available instrument, researchers can purchase or loan the instrument and reproduce the protocol outlined in the methods. However, there is no indication that the HubCentra 84 is commercially available or about to become commercially available. There is no possibility of a researcher reproducing the instrument themselves on the basis of the information provided. If the instrument is commercialized or very close to being commercially available then this should be explained in the paper. Otherwise the data contained in the paper will be of very limited interest to readers of this journal.

Major response 1: As you mentioned, the issue raised is a fundamental oversight, and your comment is indeed highly critical. Nevertheless, we are sincerely grateful for such valuable advice. When researchers initially encountered the instrument, both the manufacturer and the users were of the same nationality, and there were no obstacles to initiating the study. The company had developed an advanced machine with full automation and had begun preparing for global promotion. While the study progressed, the company revised its marketing strategy. The authors also informed the company of the necessity of providing informational materials for overseas customers. The company subsequently launched full-scale global marketing efforts, and these materials were made publicly available after the manuscript submission. Additionally, the company participated in MEDICA 2024 (https://www.medica-tradefair.com/vis/v1/en/exhprofiles/WrqNxrhMQGqfNKBfI5e90w). In response to the reviewer’s crucial comment, the latest brochure was obtained directly from the company and has been included as Supplementary Figure 1. We deeply appreciate your attention to this critical issue regarding the reproducibility of the study and thank you once again for your invaluable feedback.

Major comments 2: I find it strange that the authors did not choose to use sterile blood for preparation of the inocula of microorganisms. If you are going to make “simulated sepsis specimens”, why not include blood? This would have been relatively easy to do. The BactAlert and Bactec bottles are designed to work with blood. The speed of detection might be improved by the presence of blood. Furthermore, it is known that blood cells will release carbon dioxide in blood culture bottles and blood samples with a high white blood cell count can give false positive results. From the results of this paper, it is not known whether the presence of sterile blood would have caused false positive reactions in the HubCentra 84. At the very least, this should be highlighted as a major limitation.

Major response 2: We sincerely appreciate the reviewer’s insightful comment regarding the omission of an important element in the experimental design for hypothesis validation. The issue raised was not overlooked; initially, we intended to create "artificial septic blood samples" using horse blood to simulate human septic samples. However, due to strict regulations imposed by the Institutional Review Board (IRB) concerning the use of human blood for research purposes, we decided against this approach. Furthermore, we recognized that replicating human septic samples using animal blood inherently carries methodological limitations in terms of hypothesis validation. Thus, this approach was excluded from the final research design.

The reason for using 0.9% normal saline in the simulated sepsis specimens (SSS) in this study stems from previous findings by our research team and other studies in various countries. These studies demonstrated that clinical physicians frequently use body fluids in blood culture bottles for bacterial identification, and this process showed high identification accuracy. While it is true, as the reviewer pointed out, that the presence of blood in blood culture systems improves the real positive rate and shortens the time-to-positive, this study aimed to evaluate the system's performance in detecting non-blood samples, which are also commonly tested in real clinical settings. Normal saline was chosen for this study as it has the same osmotic concentration as body fluids, and we considered this approach more relevant to clinical research than using animal blood. Nonetheless, we acknowledge that the absence of blood represents a significant limitation. We have addressed this by including it as the first key point in the discussion's limitations section. Additionally, we have added references explaining why blood was not used and have strengthened the methodology section accordingly.

We are planning a follow-up study to determine the system's limit of detection, which will incorporate the use of blood samples. Once again, we express our gratitude for the reviewer’s thoughtful comments, which have provided valuable insights to improve our study and guide its future development.

Minor comments 1: Line 95: How does a desiccant (which is a drying agent) help to maintain anaerobic conditions?

Minor response 1: Thank you for accurately pointing out the misuse of the term. The component added to establish the anaerobic condition was not a desiccant. Instead, an oxygen-absorbing pouch was used, and the agar plate was sealed together with the pouch. The term "desiccant" has been removed and revised as follows: “GasPak™ EZ Anaerobe Pouch System with Indicator (BD), which was placed with a plate agar and enveloped within a pouch to maintain anaerobic conditions.”

Minor comments 2: Line 96: Please provide the full name for SDAC medium.

Minor response 2: I appreciate the detailed review and insightful feedback on the manuscript. SDAC is an abbreviation for Sabouraud Dextrose Agar with Chloramphenicol, and additional information has been incorporated into the text.

Minor comments 3: Line 97: The challenge set of microorganisms consisted of only 7 strains. Given that 100 or more species may be isolated in any given year from routine blood cultures it is misleading to call this selection “comprehensive”. It would have been useful to include fastidious Gram-negative bacteria such as Haemophilus influenzae or Neisseria meningitidis or Brucella species.

Minor response 3: The critique regarding the complexity of the phrasing is appreciated. The term "comprehensive" was intended to convey that the six microorganisms selected in the introduction and methods represent a broad range of clinically relevant pathogens. However, as the reviewer pointed out, it is not feasible to comprehensively include all infectious microorganisms with just six strains. Therefore, the previous statement, which could cause confusion, has been completely removed.

Minor comments 4: Line 121: “suspension” would be more accurate than “solution”.

Minor response 4: Your feedback on maintaining consistent terminology is appreciated. In the manuscript, the term referring to the liquid containing colonies was inconsistently used as "suspension" and "solution." Following the reviewer's suggestion, all instances have been standardized to "solution" throughout the text.

Minor comments 5: Line 166: “Culture bottles containing two or more CMBCS instruments…..”. This does not make sense to me.

Minor response 5: I appreciate your effort in interpreting the complex content of the manuscript. The sentence refers to drawing out all bottles with positive signals among the culture bottles tested on the same day. This clarification is necessary because at least two CMBCS devices were used to process bottles on the same date. To reduce ambiguity, the sentence has been revised as follows: “Culture bottles mounted on the same date were drawn out when a positive growth signal was detected.”

Minor comments 6: MacConkey agar is a selective medium containing bile salts and is not an optimal medium for performing colony counts of Pseudomonas. Counts of Pseudomonas are typically higher on blood agar than on MacConkey agar. There is no reason to use a selective medium when assessing the count of a pure culture.

Minor response 6: I appreciate the insightful questions raised, which demonstrate a thorough understanding of the manuscript. MacConkey agar, which contains bile salts, demonstrates selective functionality because bile acids are amphipathic molecules found in the gut that aid in fat digestion by emulsifying fats for transport in an aqueous environment and are toxic to many organisms by disrupting cellular membranes, which resemble fats. Developed in 1900, this medium underwent modifications by Albert Grunbaum and Edward Hume, evolving into its modern form by the 1930s. Modern MacConkey agar not only supports the growth of gram-negative enteric organisms but also distinguishes non-fastidious organisms such as Pseudomonas aeruginosa, which exhibits distinct colony morphology compared to Escherichia coli and Klebsiella pneumoniae. As noted in the manuscript, MacConkey agar used was a commercially available product that demonstrated typical P. aeruginosa colony morphology. Although P. aeruginosa may exhibit higher CFU counts on blood agar, MacConkey agar was chosen to ensure the selection of P. aeruginosa alone, allowing for the establishment of a pure culture. This explanation, along with appropriate references, has been added to the methods section to enhance clarity and detail.

Minor comments 7: Line 186: Please detail the statistical tests actually used.

Minor response 7: We greatly appreciate your meticulous review of the manuscript, which identified several omissions crucial to article’s completeness. The issue raised aligns with feedback from other reviewers. To address this, we complement statistical method, Wilcoxon rank-sum test, and this appropriate for nominal independent variables (automated systems) and continuous dependent variables (time-to-positive).

Round 2

Reviewer 3 Report

Comments and Suggestions for Authors

It is now apparent that the HubCentra84 is a commercially available instrument, which overcomes one of my major objections to the paper. However, I would maintain that the fact that blood was not used to evaluate a blood culture system is a very major limitation and will limit the interest of the article. Although the authors correctly point out that non-blood samples may be cultured in this way, this is not the main function or interest of the instrument.

Author Response

Comment 1. It is now apparent that the HubCentra84 is a commercially available instrument, which overcomes one of my major objections to the paper. However, I would maintain that the fact that blood was not used to evaluate a blood culture system is a very major limitation and will limit the interest of the article. Although the authors correctly point out that non-blood samples may be cultured in this way, this is not the main function or interest of the instrument.

Response 1. I agree that the major limitation of this study is the validation of the automated blood culture machine’s performance without using actual blood samples. However, we are planning a follow-up study to compare its performance using blood samples. The most critical aspect will be to develop appropriate methods for sample preparation, storage, and anticoagulation to maintain the freshness of blood samples. By doing so, we aim to fully address the reviewer’s concerns.